# Interrogating the recognition landscape of a conserved HIV-specific TCR reveals distinct bacterial peptide cross-reactivity

**Juan L Mendoza[1†], Suzanne Fischer[1], Marvin H Gee[1], Lilian H Lam[2,3], Simon Brackenridge[4], Fiona M Powrie[2,3], Michael Birnbaum[1,5], Andrew J McMichael[4], K Christopher Garcia[1,6]\*, Geraldine M Gillespie[4]\***

[1]Department of Molecular and Cellular Physiology, Stanford University School of Medicine, Stanford, United States; [2]Kennedy Institute of Rheumatology, University of Oxford, Oxford, United Kingdom; [3]Translational Gastroenterology Unit, Nuffield Department of Medicine, John Radcliffe Hospital, Oxford, United Kingdom; [4]Nuffield Department of Medicine, University of Oxford, NDM Research Building, Old Road Campus, Headington, Oxford, United Kingdom; [5]Koch Institute at MIT, Cambridge, United States; [6]Howard Hughes Medical Institute, Stanford University School of Medicine, Stanford, United States

**Abstract** T cell cross-reactivity ensures that diverse pathogen-derived epitopes encountered during a lifetime are recognized by the available TCR repertoire. A feature of cross-reactivity where previous exposure to one microbe can alter immunity to subsequent, non-related pathogens has been mainly explored for viruses. Yet cross-reactivity to additional microbes is important to consider, especially in HIV infection where gut-intestinal barrier dysfunction could facilitate T cell exposure to commensal/pathogenic microbes. Here we evaluated the cross-reactivity of a 'public', HIV-specific, CD8 T cell-derived TCR (AGA1 TCR) using MHC class I yeast display technology. Via screening of MHC-restricted libraries comprising $\sim 2 \times 10^8$ sequence-diverse peptides, AGA1 TCR specificity was mapped to a central peptide di-motif. Using the top TCR-enriched library peptides to probe the non-redundant protein database, bacterial peptides that elicited functional responses by AGA1-expressing T cells were identified. The possibility that in context-specific settings, MHC class I proteins presenting microbial peptides influence virus-specific T cell populations in vivo is discussed.

**\*For correspondence:**
kcgarcia@stanford.edu (KCG);
geraldine.gillespie@ndm.ox.ac.uk
(GMG)

**Present address:** [†]Pritzker School of Molecular Engineering and Department of Biochemistry and Molecular Biology, University of Chicago, Chicago, United States

## Introduction

Cross-reactivity represents an intrinsic feature of immunity that allows post-thymically selected T cell receptors (TCRs) to recognize distinct peptides originating from diverse microbial origins when bound by a single MHC. This considerable immune redundancy exists to cover the deficit between the predicted number of MHC-bound antigenic peptide epitopes that humans encounter during their lifetime ($\sim 10^{12}$) versus the number of unique TCRs available in the periphery ($<10^8$) (*Mason, 1998*; *Wucherpfennig, 2004*; *Sewell, 2012*). A biologically relevant consequence of cross-reactivity where infection with an initial microbe alters immunity to subsequent, non-related infecting pathogen - termed heterotypic immunity (*Gil et al., 2015*) - has been reported to influence the course of natural infections in humans (*Aslan et al., 2017*), and can also affect vaccine-mediated immunity and immune-mediated checkpoint therapy outcomes in vivo (*Sioud, 2018*). Where such cross-reactivity narrows the immune response and/or skews the enrichment of non-protective immune responses, sub-optimal immunity following infection or post-vaccination can result (*Clute et al., 2005*; *Aslan et al., 2017*). The potential to elicit immune pathology also exists, for

example, via the amplification of T cells with autoimmune potential (*Rist et al., 2009*; *ImMaDiab Study Group et al., 2018*). However, beneficial effects could also materialize if a cross-reactive response driven by the primary infecting microbe is protective against a subsequent infection involving a non-related pathogen (*Su et al., 2013*).

The possibility that cross-reactive T cells could influence immunity to non-related pathogens exists but has not been extensively explored – this is particularly relevant in the setting of HIV infection, where viral-induced gut-intestinal (GI) barrier dysbiosis could potentially allow commensal or pathogenic microbes to influence virus-specific T cell populations. Although this has been addressed to some extent in the context of HIV infection for MHC class II and CD4 T cells (*Su et al., 2013*; *Campion et al., 2014*), this topic has received only limited attention in relation to CD8+ T cells and MHC class I restricted epitopes (*Pohlmeyer et al., 2018*). To explore this, we chose a specific example corresponding to a 'public' HIV-specific TCR, namely, the AGA1 TCR, that recognizes the immunodominant, HLA-B*57:01-restricted gag-derived KAFSPEVIPMF (KF11) peptide epitope (*Stewart-Jones et al., 2012*). The KF11 peptide regularly elicits a set of closely related 'public' TCRs utilizing highly conserved TCR V alpha (AV) 5, V beta (BV) 19 and CDR3 A/B hypervariable chain motifs that were originally identified in non-related, HIV-infected B*57:01+ patients who progress slowly to AIDS (*Gillespie et al., 2006*; *Yu et al., 2007*). More recently, KF11-specific T cells utilizing near identical AGA1-related BV19-CDR3-BJ1-2 chains, or cells carrying conserved CDR3-BJ1-2 motif (x-Y-G-Y-T, where x = polar residues) segments on diverse BV chain backgrounds, have been reported in both slow and normal progressor HIV+ patients (*Mendoza et al., 2012*; *Simons et al., 2008*), suggesting that the x-Y-G-Y-T motif is frequently selected in response to the KF11 epitope. Although AGA1 TCR-mediated cross-reactive responses against broad clade variants, including C clade escape variants of KF11 have been described (*Gillespie et al., 2002*), there are no documented cases of cross reactivity against non-HIV peptides.

Using the yeast display-based approach (*Adams et al., 2011*; *Birnbaum et al., 2014a*; *Gee et al., 2018*) where ~ $2\times10^8$ random peptides were linked to and presented by HLA-B*57 on the surface of yeast cells, we interrogated the cross-reactivity of the AGA1 TCR. This strategy revealed a number of important features of AGA1 TCR-mediated specificity, including characterization of the minimal, central dipeptide motif within the 11 amino acid peptide sequences required to facilitate AGA1 TCR binding. In addition, by using TCR-retrieved peptide libraries to interrogate the non-redundant sequence databases, a number of bacterial peptides that elicited strong functional responses by AGA1-expressing T cell clones were identified. The implications of these findings are discussed.

## Results

### HLA-B57 yeast display library design and validation

Yeast cell libraries displaying HLA-B*57:03-β2m linked to a randomized peptide library were generated to screen the breadth of AGA1 TCR binding (see *Figure 1A* (i) for overview of strategy). The peptide-β2m-MHC library construct was based on previous designs (*Hansen et al., 2009*; *Birnbaum et al., 2014a*; *Gee et al., 2018*) and comprised a single peptide test reagent (KF11) or randomized peptide libraries linked to human β2m and the heavy chain of HLA-B*57:03 encoding a Y84A mutation. A myc-tag incorporated between the MHC α3 domain and the Aga2 yeast protein allows the monitoring of peptide-β2m-HLA-B*57:03 expression on the yeast cell surface (*Figure 1A* (ii)). The functionality of yeast displayed HLA-B*57:03 was confirmed initially by staining a control KF11-β2m-HLA-B*57:03 yeast display with fluorescently conjugated AGA1 TCR tetramers (*Figure 1B*). Following validation, mutagenized peptide libraries were produced. For the preparation of these libraries, peptide residues at positions 2 (Ala/Ser/Thr) and 11 (Phe/Trp/Tyr) (CΩ) were restricted to retain the preferred HLA-B*57:03 anchor binding motifs (*Barber et al., 1997*), whilst the remaining nine amino acids were randomized using degenerate NNK nucleotide addition as described previously (*Adams et al., 2011*; *Figure 1A* (ii)). The theoretical diversity was 1.06E15 unique nucleotide sequences (4.61E12 unique protein sequences), of which the experimental library contained $2 \times 10^8$ unique transformants.

To evaluate the peptide repertoire that allowed AGA1 TCR binding in the context of HLA-B*57:03, streptavidin-coated magnetic beads (MACS) conjugated to biotinylated AGA1 TCR were

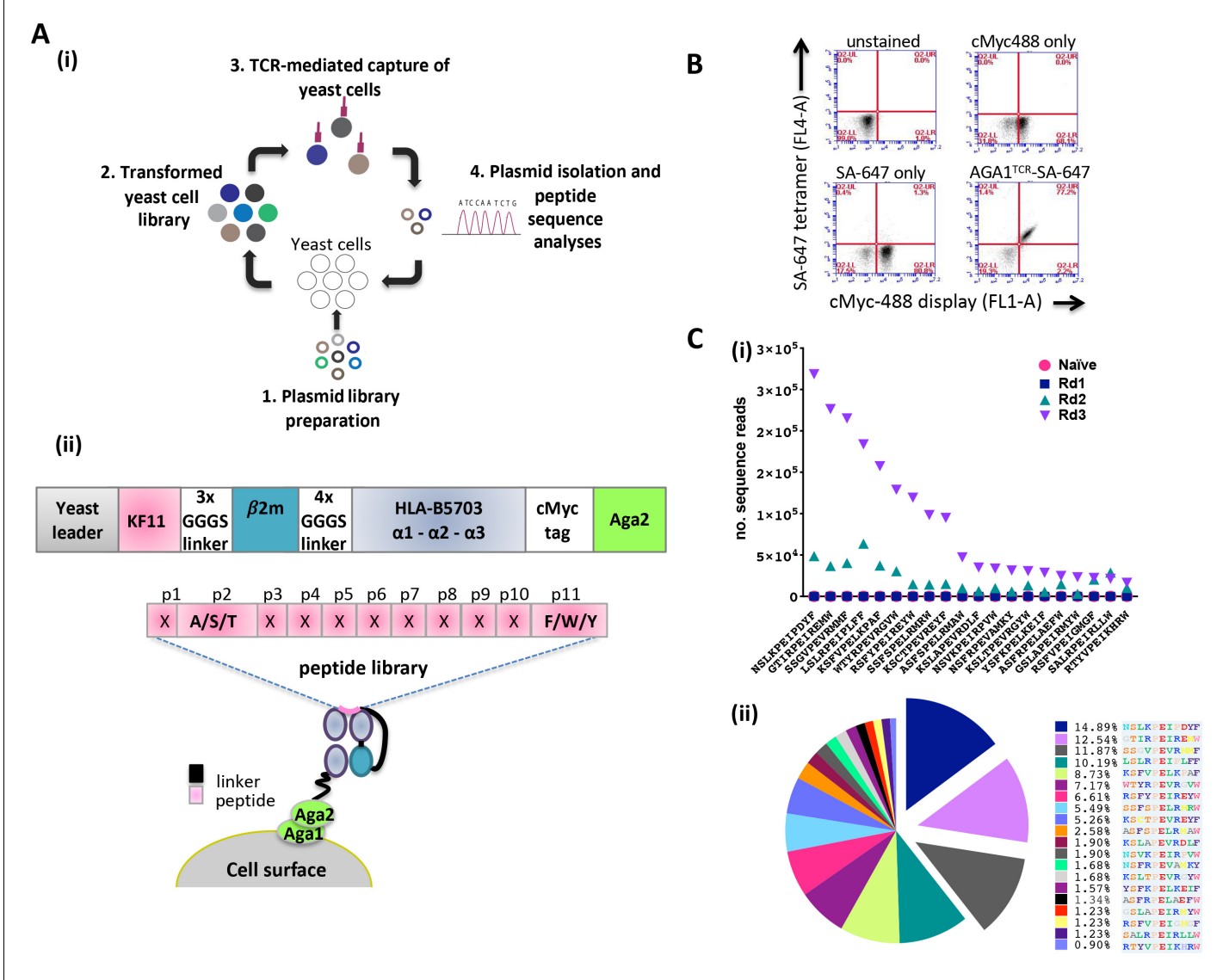

**Figure 1.** Schematic overview of the peptide-β2m-HLA-B*57:03 yeast display platform. (**A**) (i) Plasmid libraries encoding semi-randomized peptide sequences generated by polymerase chain reaction amplification using rationally designed degenerate oligos and linked to the HLA-B*57:03 heavy chain and β2m were (1) transfected into yeast cells to generate libraries (2). Biotinylated AGA1 TCR reagents conjugated to streptavidin-coated magnetic beads were subsequently used to purify TCR-reactive peptide-MHC complexes expressed on yeast cells (3). The identities of peptides allowing TCR binding were confirmed via plasmid extraction and next generation sequencing analyses (4). This process was repeated up to four times under successively stringent rounds of selection. (ii) Outline of the peptide-β2m-HLA-B*57:03-Aga2 yeast single chain construct illustrating the peptide-β2m-HLA-B*57:03 fusion chain, interspersed with repeating Gly-Ser linker sequence motifs (upper image). HLA-B*57:03-preferred anchor binding residues were fixed at position 2 (p2) to Ala/Ser/Thr and position 11 (p11, CΩ) to Phe/Trp/Tyr, whereas non-anchor residues were allowed to express any amino acid (X) throughout the selection process (lower image). (**B**) Staining of a KF11-β2m-HLA-B*57:03 yeast display test platform with AGA1 TCR fluorescent tetramers. cMyc staining denotes cell-surface expression of the KF11-β2m-HLA-B*57:03 construct, with SA-647 staining employed to monitor non-specific binding of the fluorescent label to transfected yeast cells. (**C**) Frequency of top 20 Round 3 peptide sequences, pre-(naïve) and post-AGA1 TCR-mediated selection. Individual peptide sequences (X-axis) versus their frequencies (number of sequence reads) in the naïve, Round 1, Round 2 and Round 3 AGA1 TCR-selected libraries are denoted (Y-axis) in (i), with the three dominant peptides following Round 3 enrichment illustrated in (ii) as exploded pie chart slices. Percentage (%) peptide frequencies are indicated. The corresponding peptide legend is color-coded according to RasMol amino schema.

The online version of this article includes the following source data for figure 1:

**Source data 1.** AGA1 TCR-recovered top 20 peptides_Naïve to Round 3 screens.
**Source data 2.** AGA1 TCR-recovered top 20 peptides_ Round 3 frequencies.

employed to perform iterated rounds of library selection. The enrichment of library peptides was tracked via deep sequencing analysis. By Round 3 of selection, clear evidence of peptide sequence enrichment emerged for variants that resembled the known KAFSPEVIPMF (KF11) peptide. Of the roughly $2.1 \times 10^6$ reads detected from deep sequencing runs, the top 20 selected peptides represented 89.4% of recovered reads at Round 3, and three peptide sequences accounted for approximately 35% of these reads (*Figure 1C*).

Distinct patterns of individual amino acid enrichment/fixation within the peptide sequences had also emerged by Round 3 (*Figure 2*, *Figure 2—figure supplement 1*). The most striking distribution of amino acids included the near exclusive selection of peptides with a conserved central p5Pro-p6Glu motif, which contrasted with the greater diversity tolerated at auxiliary positions along the remaining peptide sequences (*Figure 2A*). Given that the majority of interactions are formed between CDR3 alpha chain amino acids and the KF11 peptide Glu6 side chain in the published HLA*57:03-KF11-AGA1 TCR structure (*Stewart-Jones et al., 2012*; *Figure 2B*), fixation of this central motif most likely reflects its critical importance for AGA1 TCR binding. Amino acid selection outside the p5Pro-p6Glu di-motif was variable, although patterns of conservation specific to the physiochemistry of residues were apparent; the strong enrichment of hydrophobic amino acids at p3 and p7, and to a lesser extent at p10, for example, reflected the chemistries of these residues in the KF11 peptide (p3Phe, p7Val, p10Met, respectively). Similarly, p4 position - a solvent-exposed Ser and an important AGA1 TCR contact residue in the KF11 peptide - displayed a specific enrichment of polar residues in library-derived peptides. Selection patterns were less obvious at positions 1, 8 and 9, where mixtures of both polar and non-polar residues were tolerated.

## Cross-reactivity of AGA1+ T cell clones to library- and database-derived peptide hits

The top library peptide sequences from Round 3 enrichments were used as input to generate a position probability matrix, used previously to predict antigens to score peptide sequences using a sliding window along proteins from the 'non-redundant' (nr)-database containing both human and microbial proteins (*Gee et al., 2018*; *Birnbaum et al., 2014b*). From this search, six KF11-like hits were returned (*Supplementary file 1* - Table 1), implying that the yeast display selection results could uncover the cognate specificity of this TCR. However, a substantial number of microbial peptide hits that do not resemble the KF11 peptide dominated the sequence homology searches (*Supplementary file 1* -Table 2). From the top 20 peptides hits retrieved, six microbial peptide hits, (Mic 1 to 6), were selected for follow-up in T cell functional studies (Table 1); their selection was biased to include sequence from microbes potentially encountered by humans - either those associated with the environment (e.g. soil-derived), or those previously isolated from, or linked to infections in humans (*Chávez de Paz et al., 2004*; *Chhour et al., 2005*; *Wolfgang et al., 2012*). Peptides identified from the yeast display library (Lib 1 to 6) and individual 'nr'-derived microbial peptides predicted to be recognized by the AGA1 TCR were synthesized and tested for their ability to induce IFNγ secretion from an AGA1-expressing T cell clone (*Figure 3A* (i-ii) and Table 1 for the sequence identity of peptides). All responses were compared to the index KF11 gag peptide epitope. In terms of the yeast display-derived peptides, five of the six tested resulted in a measurable response. These peptides elicited IFNγ production that varied significantly across the peptide titration range, and one library peptide (Lib 2) failed to induce IFNγ production despite its selection by the AGA1 TCR in yeast display enrichment screens. Of the 'nr'-mined microbial peptides, three activated the AGA1+ T cell clone, of which two (Mic 1 and 3) peptides mapped to microbes previously linked with gingival (*Chávez de Paz et al., 2004*; *Chhour et al., 2005*) and gastrointestinal (GI) infections in humans (*Wolfgang et al., 2012*). In terms of IFNγ production, responses to these peptides approached the magnitude elicited by the index KF11 peptide. The remaining three 'nr'-derived microbial peptides drove weak or no T cell responses.

We questioned if the differential ability of the various library and microbial peptides to elicit T cell responses primarily reflected binding differences to HLA-B*57:01. This is particularly relevant for the Lib-derived peptides, which were bound to β2m as part of the HLA-B*57:03 yeast display single chain trimer construct but were presented in cellular assays as non-linked peptide epitopes in complex with HLA-B*57:01 on antigen presenting B cells. To address this, we performed a sandwich ELISA-based UV exchange peptide binding assay to gauge the ability of the various peptides to bind and therefore 'rescue' HLA-B*57:01 pre-refolded with UV-labile peptide upon photo-

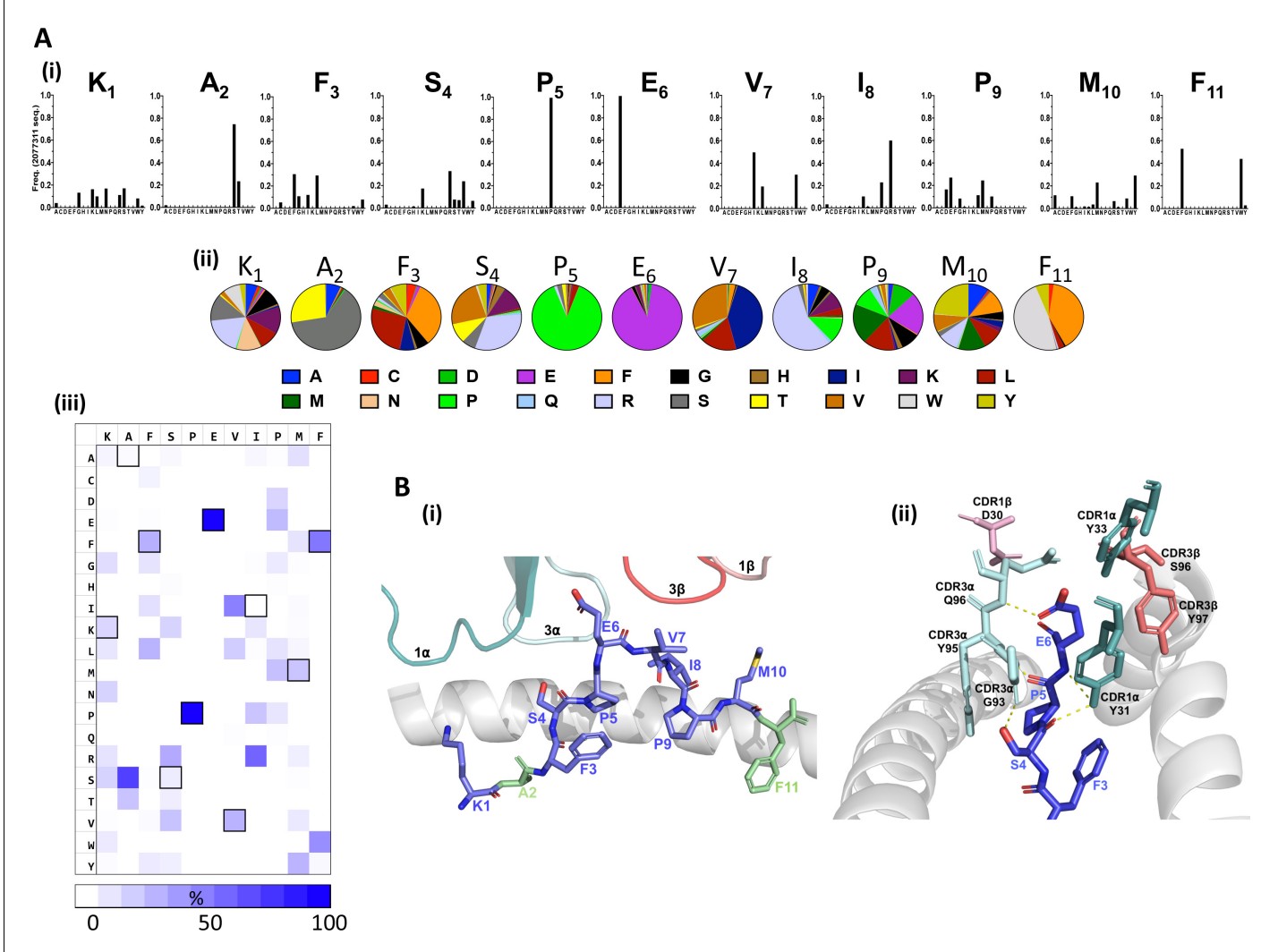

**Figure 2.** Fixation of specific peptide amino acids residues following Round 3 AGA1 TCR–mediated selection. (**A**) Amino acid enrichment along the length of AGA1 TCR selected 11mer peptides are illustrated (i) in bar chart format for approximately $2.0 \times 10^6$ selected peptide sequences, with individual amino acid enrichment at each position of the 11mer peptide represented on the X-axis and sequence frequency on the Y-axis. K1, A2, F3 refers to amino acid positions 1, 2, 3, etc, of the HIV KF11 epitope which is included for reference. Enrichment data for pre-(naïve) and post-AGA1 TCR selected peptides from all libraries are provided in *Figure 2—figure supplement 1*). (ii) The top 1000 peptide sequences are reported in pie-chart format, with the original KF11 (KAFSPEVIPMF) peptide amino acids included above the relevant pie-charts for reference. K1, A2, F3 refers to amino acid positions 1, 2, 3, etc, of the HIV KF11 peptide sequence. (iii) A Heat map of the entire Round 3 peptide sequence dataset (~$2.0 \times 10^6$ peptides) demonstrating the near absolute dominance of Pro and Glu at positions 5 and 6, respectively, in AGA1 TCR-selected peptide datasets. The amino acid residues corresponding to the KF11 peptide are outlined in black. Heatmap scale = 0 – 100%, 10% increments. (**B**) Structural overview of the primary contacts formed between the AGA1 TCR and KF11 when restricted by HLA-B*57:03. (i) Slide view of the HLA-B*57:03 alpha 1 (α1) helix in cartoon form (grey), with the alpha 2 (α2) helix removed for clarity. The peptide is depicted in stick format (blue) with the KF11 position 2 Ala (A2) and Position 11 Phe (F11) anchor resides highlighted in green. The TCR CDR1α (deep teal), CDR3α (pale cyan), CDR3β (deep salmon) and CDR1β (light pink) loops that form the main contacts with HLA-B*57:03-KF11 are illustrated. (ii) Barrel view of HLA-B*57:03-KF11, oriented from the peptide's N terminus, with HLA-B*57:03 α1 and α2 helices display in cartoon format (grey) and KF11 amino acids 3–6 illustrated in stick format (blue). The remainder of the KF11 peptide sequence is omitted for clarity. The primary polar contacts (~3 ångströms (Å)) between TCR CDR1α (deep teal) and CDR3α (pale cyan) amino acids and the KF11 peptide at positions 4 to 6 are illustrated (yellow dash lines). CDR1β residue D30 (light pink) and CDR3β amino acids S96 and Y97 (deep salmon) that reportedly form weaker peptides mediated contacts are also displayed. TCR amino acids positions are number according to Arden nomenclature (see Ref 37). Structural images were generated in PyMOL four using the Protein Data Bank coordinates 2YPL.

The online version of this article includes the following source data and figure supplement(s) for figure 2:

**Source data 1.** All Round 3 AGA1 TCR recovered library peptide amino acid frequencies.

**Source data 2.** Top 1000 Round 3 AGA1 TCR recovered library peptide amino acid frequencies.

**Source data 3.** Heatmap of all Round 3 AGA1 TCR recovered library peptides.

*Figure 2 continued on next page*

Figure 2 continued

**Figure supplement 1.** Individual amino acid frequencies in the naïve (pre-AGA1 TCR-selected) HLA-B*57:03-restricted yeast display peptide repertoire.

**Figure supplement 1—source data 1.** Amino acid signatures of Yeast display libraries-Naïve.

**Figure supplement 2.** Individual amino acid frequencies of the AGA1 TCR-selected, HLA-B*57:03-restricted yeast display peptide repertoire following Round 1 selection.

**Figure supplement 2—source data 1.** Amino acid signatures of Yeast display libraries-AGA1 TCR Round 1 selection.

**Figure supplement 3.** Individual amino acid frequencies of the AGA1 TCR-selected, HLA-B*57:03-restricted yeast display peptide repertoire following Round 2 selection.

**Figure supplement 3—source data 1.** Amino acid signatures of Yeast display libraries-AGA1 TCR Round 2 selection.

**Figure supplement 4.** Individual amino acid frequencies of the AGA1 TCR-selected, HLA-B*57:03-restricted yeast display peptide repertoire following Round 3 selection.

**Figure supplement 4—source data 1.** Amino acid signatures of Yeast display libraries-AGA1 TCR Round 3 selection.

**Figure supplement 5.** Individual amino acid frequencies of the AGA1 TCR-selected, HLA-B*57:03-restricted yeast display peptide repertoire following Round 4 selection.

**Figure supplement 5—source data 1.** Amino acid signatures of Yeast display libraries-AGA1 TCR Round 3 selection.

illumination. In terms of the library-derived peptides, there was a strong concordance between the magnitude of T cell responses generated and the extent to which the peptides stabilised HLA-B*57:01 (*Figure 3B*). Lib 2 peptide, for example, demonstrated the weakest binding to HLA-B*57:01 and represented the only library-derived peptide that failed to elicit functional responses by AGA1-expressing T cell clones. A similar pattern existed for the microbial peptides, where the largest T cell responses were generated by the strongest HLA-B*57:01 binding peptides. Interestingly, four peptides exhibited enhanced binding relative to the index KF11 peptide, including the Mic 1 peptide, which corresponds to a *Sporosarcina newyorkensis*-derived halodehydrogenase (HdH) peptide, and its closest library sequence match, Lib 1.

We re-analyzed responses to the database-derived microbial peptides using a second T cell clone isolated from a different donor that expressed an AGA1-homologous TCR, namely clone 1.2 (*Figure 3C* and *Supplementary file 1* - Table 3). The functional screens were extended to assess both IFNγ production and CD107a up-regulation in response to the microbial peptides. This T cell clone also demonstrated a clear preference for the 'nr'-mined *Olsenella uli*, *Candida orthopsilosis* peptides and in particular, the *S. newyorkensis* peptide, both in terms of IFNγ production and up-regulation of the LAMP protein, CD107a (*Figure 3C* (i) and (ii), respectively).

## Recognition HdH peptides by AGA1-expressing T cell clones

As the Mic 1 peptide, which mapped to the haloacid dehydrogenase (HdH) enzyme of *S. newyorkensis,* elicited the strongest T cell-mediated responses at concentrations comparable to the KF11 epitope, we chose this peptide for follow-up studies. We re-interrogated the 'nr' database using this peptide motif to hunt for related peptides. A number of highly homologous peptides were identified of which the majority mapped to HdH enzymes specific to distinct microbial genera. Ten additional HdH peptides (Table 2) were synthesized and tested in T cell functional assays. The majority of these peptides elicited IFNγ production at titrations comparable to the *S. newyorkensis* (HdH1/Mic1) peptide, and only marginally weaker than responses induced by the index KF11 epitope (*Figure 4A*). One peptide (HdH2), was recognized at the highest peptide concentration only, whereas two additional peptides (HdH5 and HdH6) failed to elicit T cell responses. To address if this related to HLA-B*57:01 binding or recognition by the AGA1 TCR, we tested all HdH peptides in the sandwich ELISA-based UV exchange peptide binding assay (*Figure 4B*). When the HdH and KF11 peptides were ranked according to the strength of peptide binding ELISA data, specific amino acids were prominent in the stronger binding peptides: position (p)2 Thr/Ser (preferred anchors), p3Iso, p7Iso, a Pro at p8 or 9 (but not both) and p10Try. Peptides in the medium- and low-binding categories, although incorporating some of these residues, had either single or combined differences at these positions (*Figure 4B* and *Supplementary file 1* - Table 4). We also compared UV peptide-exchange peptide binding ELISA data to NetMHCPan4.1 (http://www.cbs.dtu.dk/services/NetMHCpan/) predicted binding affinities for KF11 and the HdH peptides (*Figure 4*, *Figure 4—figure supplement 1*). Although a negative correlation was observed, this association was very weak (R = −0.18), and most

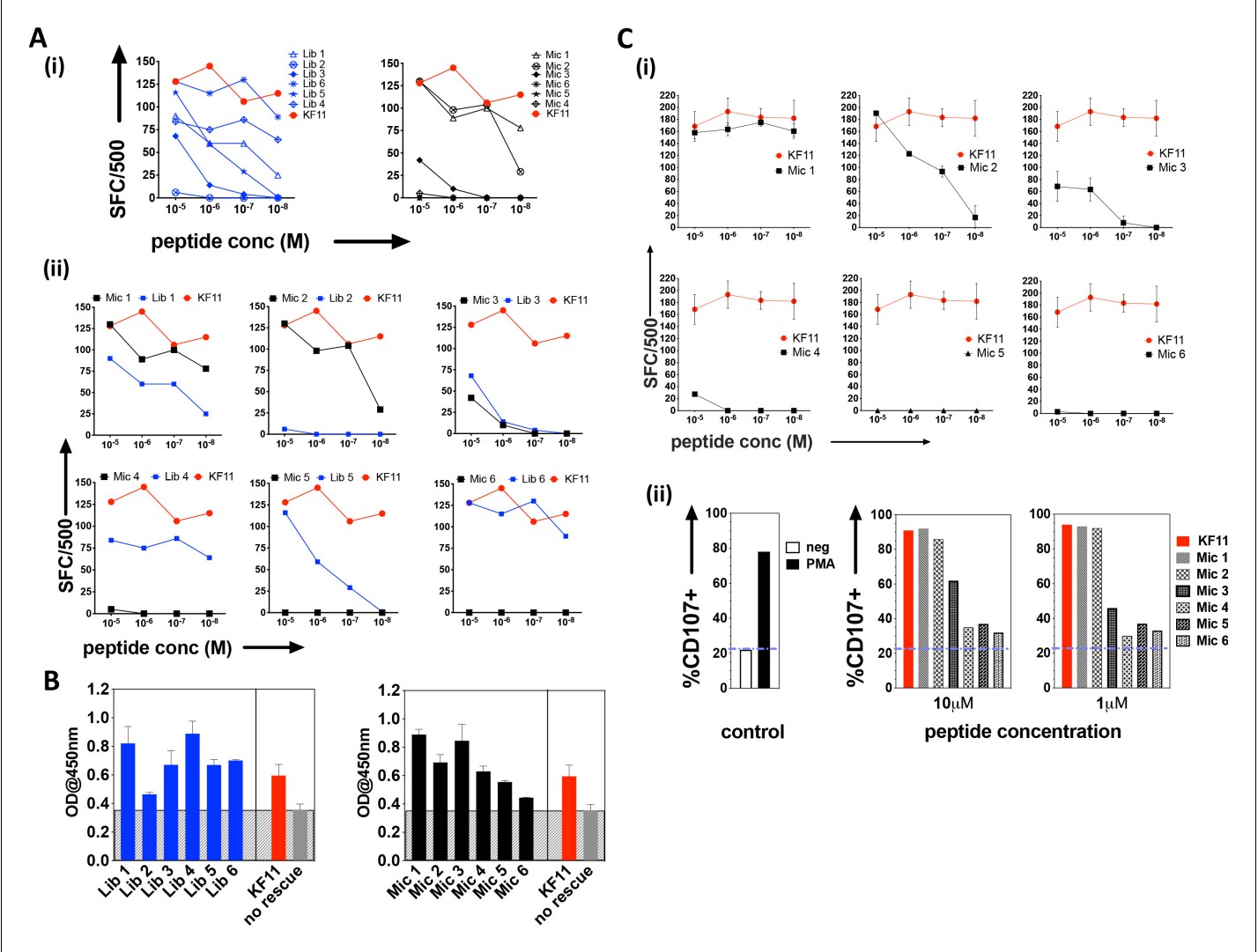

**Figure 3.** Recognition of library-derived peptides and their closest peptide sequence matches identified from the 'nr' database by AGA1-expressing T cell clones. (A) Recognition of six library peptides (Lib) and their closest 'nr'-derived microbial peptides (Mic) matches were compared to the index KF11 in IFN-γ based ELISpot assay screens using an AGA1-expressing T cell clone, summarized in (i) with individual Lib and Mic peptide responses plus comparison to KF11 denoted in (ii). Peptide concentration (Molar (M)) is denoted on the X-axis, with numbers of Spot Forming Cells (SFC) per 500 cell input on the Y-axis. Initial screens were performed with one replica per screen, and were screened on two separate occasions following re-stimulation of T cell clone 1.1 (biological repeat, n = 2), with one representative shown here. Test peptide IDs, their origins and sequence identities are specified in Table 1. (B) Binding of library derived and the sequence-mined microbial peptides to HLA-B*57:01, assessed by UV-mediated peptide exchange sandwich ELISA. The Y-axis denotes average absorbance readings at 450 nm, with the peptides tested reported on the X-axis. The background corresponding to the no peptide rescue (nr) control is denoted in grey (also illustrated across the samples by grey hatching). Assays were performed in duplicate (technical repeats, n = 2) on two separate occasions using different peptide stock dilutions (biological repeats, n = 2), with one representative shown here. Error bars corresponding to the Standard Error of the Mean (SEM) are reported. Test peptide IDs, their origins and sequence identities are specified in Table 1. (C) Recognition of the six nr-derived microbial peptides (Mic 1–6) by the closely related AGA1-like T cell clone 1.2, assessed by IFN-γ based ELISpot assay and summarized in (i) with comparison to the KF11 index peptide reported. Peptide concentration (Molar (M)) is denoted on the X-axis with the numbers of Spot Forming Cells (SFC) per 500 cell input on the Y axis. Assays were performed in duplicate (technical repeats, n = 2) on two separate occasions using different peptide stock dilutions and following re-stimulation and resting of T cell clone 1.2 (biological repeats, n = 2), with one representative shown here. Error bars corresponding to the Standard Error of the Mean (SEM) are reported. Test peptide IDs, origins and sequences are specified in Table 1. (ii) Up-regulation of CD107 on T cell clones in response to 1 and 10 µM Mic peptides compared to the index KF11 epitope was assessed by flow cytometry. Negative (no added peptide - purple dashed line) and positive controls (10 ng/mL PMA stimulation) are reported for reference.

The online version of this article includes the following source data for figure 3:

**Source data 1.** AGA1+ T cell clone 1.1 recognition of top 6 Lib and closest 'nr' mined microbial (Mic) peptide hits_all combined_ELISpot data.
**Source data 2.** AGA1+ T cell clone 1.1 recognition of top 6 Lib and closest 'nr' mined microbial (Mic) peptide hits_all combined_ELISpot data.
**Source data 3.** UV-exchange HLA-B*57:01 peptide binding ELISA data for Top 6 Lib and closest 'nr' mined microbial (Mic) peptides.

**Source data 4.** AGA1+ T cell clone 1.2 recognition of 6 'nr' mined microbial (Mic) peptide hits_individual plots_ELISpot data.

**Source data 5.** AGA1+ T cell clone 1.2 recognition of 6 'nr' mined microbial (Mic) peptide hits_CD107 data.

likely reflects a general lack of data-informed algorithm training for MHC class I with longer peptide using artificial neural network analysis-based systems such as NetMHC.

In terms of the individual peptides, HdH6 demonstrated weaker HLA-B*57:01 binding, potentially owing to the presence of a non-favored (Glu) anchor residues at p2. In contrast, HdH2 and HdH5 produced strong ELISA binding signals, and their impact was potentially mediated at the level of TCR recognition, either via interactions that directly abrogate the peptide's central 'p5Pro-p6Glu' TCR recognition hotspot (via replacement of p5Pro with Glu in HdH5, for example) or mutations distal to the hotspot that potentially distorts the normal trajectory of the peptide required to facilitate AGA1 TCR binding (p9Pro to Ser mutation in HdH2).

To compare the overall sequence commonality between the canonical KF11 peptide, the top 20 library-derived peptides and the panel of HdH peptides recognized by the AGA1-expressing T cell clones, Seq2Logo motifs were generated. As illustrated in *Figure 4C*, the AGA1 TCR-selected yeast display peptide library logo reflected tolerance to a broad array of amino acids beyond the conserved, central dipeptide 'p5Pro-p6Glu' motif. The HdH-based logo allowed further refinement of the motif based on T cell functionality and demonstrated that enhanced recognition by AGA1 TCR-expressing T cell clones required further fixation of amino acids outside the central 'dipeptide' motif; most notably, with a preference for p9Pro in HdH peptides. This logo also illustrated that despite low KF11 sequence homology (40–50%), the majority of HdH peptides elicited functional T cell responses at magnitudes approaching that generated by the KF11 peptide.

## Recognition of processed antigens

Although antigen presenting cells pulsed with exogenous 'nr' mined microbial peptides elicited strong cross-reactive T cell responses, we next wanted to ascertain if T cell-mediated recognition of these peptides occurred under conditions that more accurately reflect the physiological processing of microbial antigens in vivo. To test this, we generated lysates of the three microbial species that elicited AGA1 TCR-specific responses in the peptide-based functional assays. The lysates were incubated with HLA-B*57:01-expressing HL60 cells (*Zorn et al., 2002*) pre-treated overnight with calcium ionophore and IFNγ to induce DC-myeloid-like cellular differentiation and enhance MHC class I expression (*Wu et al., 2004a*). Following 7 hr of incubation, the cells were washed extensively and their ability to stimulate AGA1-expressing clone 1.1 and the related clone, 1.2, was assessed by ELISpot to evaluate IFNγ secretion. As demonstrated in *Figure 4D*, T cell clones 1.1 and 1.2 specifically responded to lysates generated from *S. newyorkensis* whereas reactivities to the other lysates, including the negative control *Ruminococcus gnavus* lysate, were not detected.

## Discussion

Cross-reactivity represents an essential component of T cell-mediated immunity that allows a relatively small repertoire of TCRs to mount effective immune responses against the larger number of pathogenic peptide epitopes encountered during a lifetime. A physiologically relevant outcome of cross-reactivity, where previous exposure to one microbe alters the immune response to a subsequent, non-related pathogen (heterotypic immunity) has the potential to drive either efficacious or pathogenic outcomes in vivo (*Gil et al., 2015*). This interesting phenomenon has been almost exclusively explored in the context of viral immune responses, and whether peptides of non-viral origin, such as those originating from bacteria, yeast or fungi can influence viral-specific CD8+ T cell immunity by mechanisms involving MHC class I restriction has received limited exploration (*Pohlmeyer et al., 2018*). This is especially relevant in the setting of HIV infection, where viral-induced gut-intestinal (GI) barrier dysfunction (*Brenchley et al., 2006a*; *Brenchley et al., 2006b*) and microbiome dysbiosis (*Dillon et al., 2016*; *Mudd and Brenchley, 2016*; *Nowak et al., 2015*; *Lozupone et al., 2013*) could potentially allow commensal or pathogenic microbes to influence the frequencies/functions of resident or infiltrating virus-specific T cells.

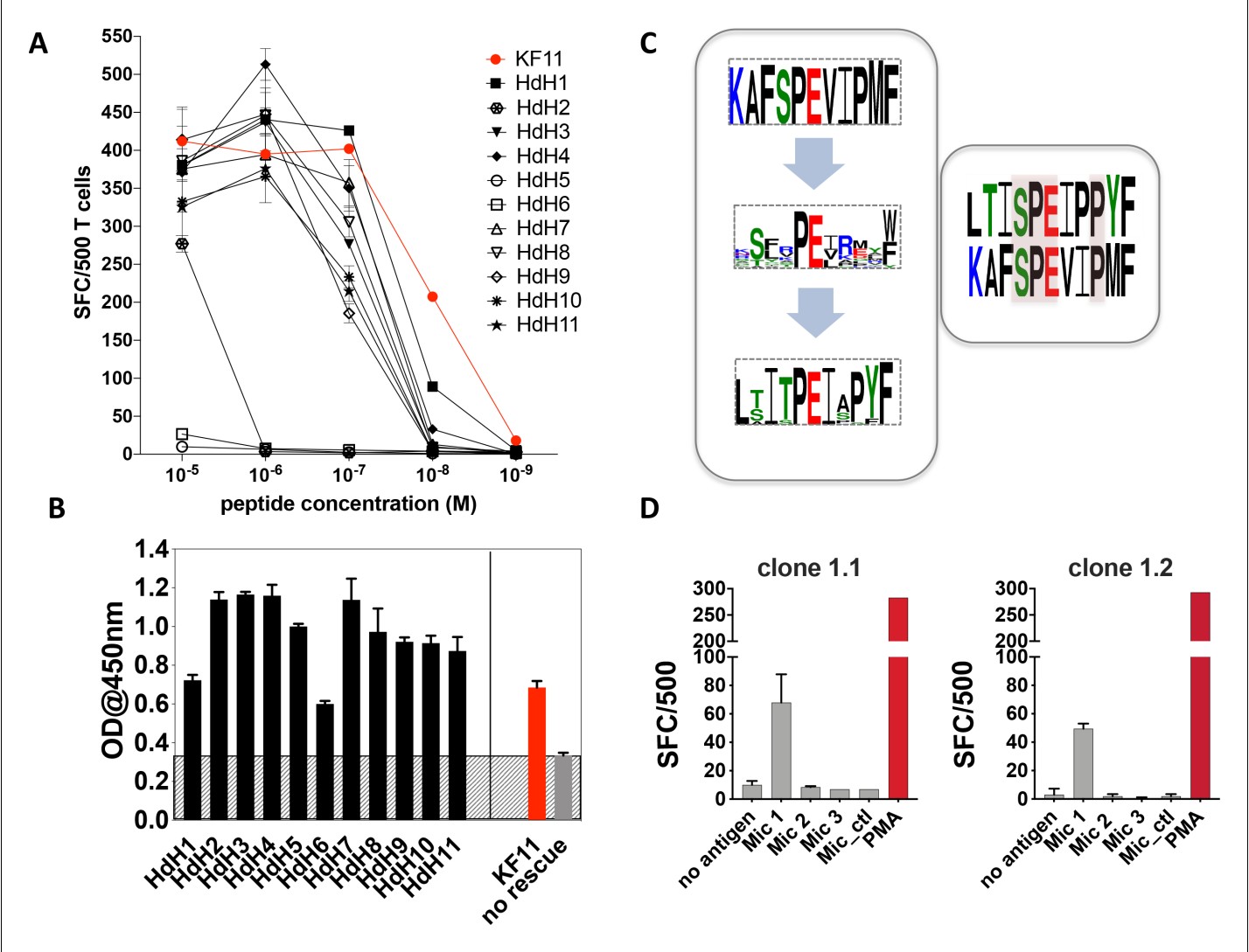

**Figure 4.** Recognition of haloacid dehydrogenase (HdH) peptides and bacterial lysate-derived antigen by AGA1-expressing T cell clones. (**A**) Ten non-redundant ('nr') database-mined HdH peptides with close homology to *S. newyorkensis* peptide (Mic 1/HdH1) were tested in an IFN-γ based ELISpot assays using an AGA1-expressing T cell clone. Peptide concentration (Molar (M)) is denoted on the X-axis and the numbers of Spot Forming Cells (SFC) per 500 cell input is displayed on the Y-axis. Responses to the KF11 epitope are noted in red. Assays were performed in duplicate (technical repeats) on two separate occasions using different peptide stock dilutions and following re-stimulation and resting of T cell clone 1.2 (biological repeats), with one representative shown here. Error bars corresponding to the Standard Error of the Mean (SEM) reported. Test peptide IDs, their origins and sequence identities are specified in Table 2. (**B**) Binding of the HdH peptides to HLA-B*57:01, assessed in the UV-mediated peptide exchange sandwich ELISA assay. The Y-axis denotes average absorbance readings at 450 nm and the X-axis denotes test peptides. The background corresponding to the no peptide rescue (nr) control is denoted in grey (also illustrated across the samples by grey hatching). Assays were performed in duplicate (technical repeats) on two separate occasions using different peptide stock dilutions (biological repeats, n = 2), with one representative shown here. Error bars corresponding to the Standard Error of the Mean (SEM) are reported. Test peptide IDs, their origins and sequence identities are specified in Table 2. (**C**) Summary of evolved peptide motifs recognised by the AGA1 TCR. Recognition beyond the original KAFSPEVIPMF (KF11) index motif is exemplified initially by the diverse peptide sequences retrieved during repeated rounds of AGA1 TCR-mediated peptide selection, with a Seq2Logo motif reported for the top 20 Round 3 evolved peptide libraries. Following evaluation of 'nr'-database derived peptides in T cell functional assays, peptides that elicited the strongest functional responses - in this case, a *S. newyorkensis*-derived haloacid dehydrogenase peptide -allowed further refinement of database-led search motifs and identification of related peptide that were functionally recognized by AGA1 TCR-expressing T cell clones. Amino acids shared between KF11 and the *S. newyorkensis*-derived Mic1/HdH1 peptide is illustrated in the smaller right panel (pink shading). (**D**) Recognition of bacterial cell lysates from *S. newyorkensis* (Mic 1), *C. orthopsilosis* (Mic 2), *O. uli* (Mic 3) and *R. gnavus* (control) by AGA1-expressing T cell clones 1.1 and 1.2 was tested using an IFN-γ based ELISpot assay. Bacterial cell lysates (20μg/mL) were incubated with cytokine-matured HLA-B*57:01 positive HL60 cells for 7 hours, following which T cell responses were evaluated. PMA (10ng/mL) was included as a positive control, and the background control comprised HL60 cells incubated with T cells only. Lysate identity is denoted on the X-axes and the numbers of Spot Forming Cells (SFC) per 500 cell

*Figure 4 continued on next page*

*Figure 4 continued*

input are displayed on the Y-axes. Assays were performed in duplicate (technical repeats, n = 2) on two separate occasions (biological repeats, n = 2) using fresh lysate stock dilutions and following re-stimulation and resting of T cell clones 1.1 and 1.2. One representative is shown.

The online version of this article includes the following source data and figure supplement(s) for figure 4:

**Source data 1.** AGA1+ T cell clone 1.2 recognition of 'nr' mined HdH peptides_ELISpot data.
**Source data 2.** UV-exchange HLA-B*57:01 peptide binding ELISA data for HdH peptides.
**Source data 3.** Recognition of S. newyorkensis bacterial lysates by AGA1+ T cell clones 1.1 and 1.2_ELISpot data.
**Figure supplement 1.** Weak, negative correlation between HLA-B*57:01 UV exchange-peptide binding data and Net MHCpan 4.1 predicted affinities for KF11 and the HdH peptides.
**Figure supplement 1—source data 1.** Correlation between UV exchange HLA-B*57:01 peptide binding data and NetMHC pan4.1 predicitons.

To explore cross-reactivity beyond the normal scope of viral-derived peptide epitopes we chose the well-characterized and commonly selected HLA-B*57:01-restricted AGA1 TCR that recognizes the HIV gag-derived KF11 epitope. Although specific features, including the short and convergent nature of both the TCR CDR3alpha/beta regions (*Venturi et al., 2006*), in addition to the unusual nature of the KF11 peptide when bound by HLA-B*57:01 (*Stewart-Jones et al., 2005*), may drive preferential usage of this TCR, external influences could additionally contribute to its enhanced frequency or common selection in vivo. To evaluate the cross-reactive breadth of the AGA1 TCR, we employed previously developed MHC class I yeast display technology (*Adams et al., 2011*; *Birnbaum et al., 2014a*; *Gee et al., 2018*). This approach permits a high-throughput, non-biased screen of cross-reactivity via TCR-mediated sampling of a repertoire approximating ~$2 \times 10^8$ distinct peptides bound by MHC on the yeast cell surface. Data obtained using this well-validated approach illustrated key features of AGA1 TCR-mediated specificity. The first and most notable included fixation of the peptide's central p5Pro-p6Glu di-motifs in TCR-selected libraries following 3 rounds of yeast display selection. As illustrated by our previous analysis of the HLA-B*57:03-KF11-AGA1 TCR co-crystal complex, KF11 peptide residues 4 to 6 primarily contribute to extensive H bonding and van der Waals interactions, the majority of which are formed between CDR3 alpha chain amino acids and the KF11 peptide's p6Glu residue (*Stewart-Jones et al., 2012*). The importance of p6Glu for AGA1 TCR-mediated recognition of library peptides was also reflected here by its absolute conservation in peptide sequences retrieved following repeated rounds of selection. The near equivalent requirement for p5 Pro in library-enriched peptides may instead reflect its indirect role facilitating the precise peptide conformation to generate a preferred AGA1 TCR recognition landscape akin to that described for KF11 in complex with HLA-B*57:03. Despite the substantial involvement of KF11 p4Ser in the published AGA1 TCR-B*57:03-KF11 binding interface, this residue was not completely conserved in all peptides following selection. However, as library peptides with p4 polar residues were specifically enriched, this may reflect pressure to fulfil H bonding requirements analogous to those described previously for the interaction between KF11p4Ser and the AGA1 TCR CDR3 alpha chain region in the AGA1 TCR-B*57:03-KF11 co-complex structure (*Stewart-Jones et al., 2012*).

Although the library design generated in this study allowed for the KF11 sequence to be present, it was not found in either the naïve or in subsequent rounds of AGA1 TCR-enriched libraries. Hence, the yeast clone was unlikely to be a transformant in these screens. This, to some extent, reflects one small limitation of the yeast display screens where the functional library cannot fully cover the theoretical diversity of all potential cross-reactive peptides. However, the real strength of the method is reflected by its ability to identify the restricting peptide when absent, in addition to related peptides, from available databases using algorithms and statistical-based approaches. This is what we observed for the KF11 peptide which emerged as the top database-mined peptide hit using library peptide-led database screens.

Following Round 3 peptide sequence acquisition, data-informed algorithms were developed to identify similar peptides in the non-redundant database. Next to KF11-mined peptides and related variants, microbial peptides comprised the top peptide hits. Of the small panel of bacterial peptides included in follow-up T cell functional studies, three peptides were recognised. One peptide that elicited weak responses was derived from *O. uli*, a gram-positive member of the *Coriobacteriaceae* family previously associated with orthodontic infections in humans (*Vieira Colombo et al., 2016*; *Dewhirst et al., 2001*). A second peptide mapped to a hypothetical protein (CORT_0A05310) from *C. orthopsilosis*, an invasive yeast strain associated primarily with blood-borne infections linked to

catheter-delivered nutrients (hyperalimentation) during hospitalization (*Tavanti et al., 2005*; *Yong et al., 2008*; *Silva et al., 2009*; *Merseguel et al., 2015*). However, the largest responses were driven by a peptide sequence (LTISPEIPPYF) that mapped to a haloacid dehydrogenase (HdH) enzyme from *S. newyorkensis*. Although normally associated with terrestrial environments and food production/processing facilities, these endospore-forming Firmicutes have also been isolated from plants, in addition to humans and animals specimens (blood and fecal) (*Oliver et al., 2018*), (*Wolfgang et al., 2012*), most likely as a consequence of exposure to contaminated soil. Although KF11 and the *S. newyorkensis*-derived HdH peptide share only 50% sequence homology, keys amino acids were conserved; these included peptide amino acids comprising a central TCR recognition 'hotspot' (p4Ser and p6 Glu) in addition to residues that shaped the solvent-exposed peptide arch (p5Pro and p9Pro) that helped both stabilize interactions with HLA-B*57:01 and allowed AGA1 TCR binding (*Stewart-Jones et al., 2012*). Through further rounds of database interrogation, a panel of closely related HdH peptide variants were identified from distinct bacterial genera, of which the majority were also recognised by AGA1-expressing T cell clones. Interestingly, one of these peptides (LAITPEIAPYF) mapped to *Bacillus massiliosenegalensis*, a species previously isolated from the gut microbiome in humans (*Forster et al., 2019*; *Ramasamy et al., 2013*). Collectively, our findings raise the question that if encountered in a HLA-B*57:01-restricted context, could haloacid dehydrogenase peptides originating from these diverse microbes influence the frequencies and/or functionality of AGA1 TCR expressing T cells in vivo?

It is known that certain bacteria, either directly via bacterial-derived LPS (*Tripathy et al., 2017*), CpG DNA (*Speiser et al., 2005*) and metabolites generated by specific microbiome-resident populations (*Luu et al., 2018*) or indirectly, via the induction of cytokine expression (*Wong and Pamer, 2001*), can affect the functionality or non-specifically drive the expansion of CD8+ T cells. Whether virus-specific CD8+ T cell populations are occasionally expanded by bacterial-derived antigens in an MHC-restricted context is less well understood. Although this question remains unanswered here for the AGA1 TCR, CD8+ T cells that specifically cross-react with both self-peptide and commensal microbes have been reported, mainly in the context of autoimmune disease. In the murine NOD MyD88-/- model of type I diabetes, the disease-associated enrichment of fusobacteria encoding a peptide with homology to the islet-specific glucose-6-phosphate catalytic subunit-relate (IGRP) protein, reportedly drove activation of IGRP-reactive CD8+ T cells that promoted development of diabetes (*Tai et al., 2016*). A subset of autoreactive CD8+ T cells with specificity for an islet-derived peptide epitope that cross-reacts with a peptide derived from the commensal organism *Bacteroides stercoris*, has also been reported in human type I diabetic patients (*ImMaDiab Study Group et al., 2018*). Evidence that virus-specific CD8+ T cell frequencies are influenced by bacterial-derived peptides is less clear. In a murine study, elevated numbers of lung-derived MCMV-specific CD8 T cells was attributed to indigenous microbiota (*Sprent et al., 2000*), and although subclinical MCMV reactivation in lung tissue as a contributing factor was not fully excluded, heterologous recognition of peptides from endogenous microbes sharing consensus motifs with MCMV was proposed as a potential driver of these cells.

HIV-specific CD8+ T cells are abundant in the GI tract (*Ferre et al., 2010*), an environment that represents a major site of HIV replication and reservoir of viral persistence, thus providing a constant source of cognate antigen in vivo. Hence, the extent, if any, to which bacterial-derived MHC class I restricted peptides would influence HIV-reactive T cell frequencies in this environment is unclear. However, T cells infiltrating the Lamina Propria and intestinal intraepithelial encounter diverse pathogenic and commensal microbes in healthy donors (*Isakov et al., 2009*). With exposure to bacterial antigen exacerbated in HIV infection via translocation and systemic dissemination of microbial products following mucosal barrier disruption, it remains unknown if this setting allows specific HIV-reactive CD8+ T cell populations to be influenced by microbes in an MHC-restricted context, and specifically, if MHC class I proteins can present microbial peptides that inflate specific subsets of memory HIV-specific T cell populations during active disease. Additionally, determining if specific circumstances, for example, during gastrointestinal infections in healthy individuals or in patients with chronic Inflammatory Bowel Disease, allow MHC-restricted microbial peptides to influence the precursor frequencies of viral-specific CD8+ T cells would be of interest. Although proposed in this study, definitive evidence that specific bacteria carrying KF11-related HdH peptides pre-select and expand AGA1-expressing CD8 positive T cells in vivo is lacking, hence further studies are required to investigate this hypothesis.

# Materials and methods

## Key resources table

| Reagent type (species) or resource | Designation | Source or reference | Identifiers | Additional information |
|---|---|---|---|---|
| Cell line (*S. cerevisiae*) | EBY100 | D. Wittrup, PMID:9181578 | | MATa AGA1::GAL1-AGA1::URA3 ura3-52 trp1 leu2-delta200 his3-delta200 pep4::HIS3 prbd1. 6R can1 GAL |
| Strain, strain background (*Escherichia coli*) | Rosetta 2 | Novagen | 71400–3 | F⁻ *ompT hsdS*$_B$(r$_B^-$ m$_B^-$) *gal dcm* (DE3) pRARE2 (Cam$^R$) |
| Genetic reagent (*Homo sapiens*) | AGA1 TCR biotin tag, PET22b+ construct | This paper | | disulfide bridge was introduced into both the alpha and beta TCR chains, hexahistidine tag and a BirA biotinylation substrate motif |
| Recombinant DNA reagent | Peptide-HLA-B*5703, PYAL vector | This paper | | a single chain fusion display where HLA-B*57:03 heavy chain domains were included downstream of the KF11 epitope sequence and the β2m gene |
| Commercial assay or kit | streptavidin microbeads | Miltenyi Biotec | 130-048-102 | |
| Commercial assay or kit | LS column | Miltenyi Biotec | 130-042-401 | |
| Commercial assay or kit | Zymoprep II kit | Zymo Research | D2004 | |
| Commercial assay or kit | MiSeq Reagent V2 kit | Illumina | MS-102–2002 | |
| Antibody | anti-myc 488 (mouse monoclonal, 9B11) | Cell Signaling | 2279 RRID:AB_2151849 | |
| Peptide, recombinant protein | Streptavidin Alexa 647 | PMID:17187819 | | |
| Software, algorithm | PandaSeq | PMID:22333067 | | |
| Software, algorithm | Geneious, version 6 | Biomatters, Inc | | |
| Software, algorithm | Perl scripts for data prep, analysis and prediction | PMID:24855945 | | |
| Software, algorithm | Matlab, R2015b | Mathworks, Inc | | |
| Cell line (*Homo sapiens*) | HL60 | ATCC | RRID:CVCL_0002 | Antigen processing experiments |
| Antibody | Anti-human CD8-APC (mouse monoclonal, clone RPA-T8) | BD Bioscience | 555369 RRID:AB_398595 | Functional T cell assay (2.5 µL/test) |
| Antibody | Anti-human CD107a PE (mouse monoclonal, clone H4A3) | BD Bioscience | 555801 RRID:AB_396135 | Functional T cell assay (3 µL/test) |
| Chemical compound, drug | BD Golgi Stop | BD Bioscience | 554724 | Protein transport inhibitor for T cell functional studies (1:1500) |
| Chemical compound, drug | BD Cytofix | BD Bioscience | 554655 | Cell fixation prior to flow cytometric analysis (150 µL/tube) |

*Continued on next page*

*Continued*

| Reagent type (species) or resource | Designation | Source or reference | Identifiers | Additional information |
|---|---|---|---|---|
| Software, algorithm | FlowJo | FlowJo-BD | | Analysis of flow cytometry data |
| Commercial assay or kit | Human IFNγ ELISpot[PLUS] kit-ALP strips | MABTECH | 3420-4AST | T cell functional ELISpot assays to assess IFN-γ production. |
| Software, algorithm | AID ELISpot Classic | resolving IMAGES | | |
| Peptide, recombinant protein | synthetic peptides | This paper | KF11, HdH1-11, Mic1-6, Lib1-6 | Synthetic KF11, Mic, Lib and HdH peptides for functional T cell studies – as per concentrations and titrations in Materials and methods. |
| Others | Interferon γ | PeproTech | 300-02-100µg | Maturation of antigen presenting cells (1000 U/mL) |
| Chemical compound, drug | Calcium Ionophore | Sigma | A23187 | Maturation of antigen presenting cells (100 ng/mL) |
| Chemical compound, drug | PMA | Sigma | P1585 | Positive control for T cell functional assays (10 ng/mL) |
| Strain, strain background (*Sporosarcina newyorkensis*) | *Sporosarcina newyorkensis* | DSMZ-German Collection of Microorganisms and Cell Cultures GmbH | DSM-23544 | Bacterial cultures for generation of lysates (20 µg/mL) |
| Strain, strain background (*Olsenella uli*) | *Olsenella uli* | DSMZ-German Collection of Microorganisms and Cell Cultures GmbH | DSM 7084 | Bacterial cultures for generation of lysates (20 µg/mL) |
| Strain, strain background (*Candida orthopsilosis*) | *Candida orthopsilosis* | DSMZ-German Collection of Microorganisms and Cell Cultures GmbH | DSM 24508 | Bacterial cultures for generation of lysates (20 µg/mL) |
| Strain, strain background (*Ruminococcus gnavus*) | *Ruminococcus gnavus* | DSMZ-German Collection of Microorganisms and Cell Cultures GmbH | CC55_001C | Bacterial cultures for generation of lysates (20 µg/mL) |
| Commercial assay, kit | Pierce BCA Protein Assay Kit | Thermo Fisher Scientific | 23225 | Quantification of bacterial cell lysates. |
| Genetic reagent (*Homo sapiens*) | HLA-B*57:01 biotin tagged | PMID:11953462 | | Expression plasmid for HLA-B*57:01 protein expression. |
| Peptide, recombinant protein | 9MT4 UV peptide | PMID:21430058 | | UV exchange peptide for HLA-B*57:01 protein refold (1:100) |
| Antibody | anti-human ABC (mouse monoclonal, clone W6/32) | Biolegend | 311402 | Coating antibody for UV-peptide exchange ELISA (10 µg/mL) |
| Antibody | anti-human B2M biotin (mouse, monoclonal, clone 2M2) | Biolegend | 316308 RRID:AB_493689 | Detection antibody for UV-peptide exchange ELISA (1 µg/mL) |
| Chemical compound, drug | ExtrAvidin-peroxidase | Sigma | E2886 | Assay development step (1) for UV-peptide exchange ELISA (1:1000) |

*Continued on next page*

*Continued*

| Reagent type (species) or resource | Designation | Source or reference | Identifiers | Additional information |
|---|---|---|---|---|
| Chemical compound, drug | TMB High sensitivity Substrate | Biolegend | 421501 | Assay development step (2) for UV-peptide exchange ELISA (100 µL/well) |
| Chemical compound, drug | Stop Solution | Biolegend | 423001 | Assay development stop step (3) for UV-peptide exchange ELISA (100 µL/well) |
| Software, algorithm | FLUOstar | BMG LABTECH | | ELISA plate absorbance reading@450 nm |

## Design and construction of HLA-B*57:03 yeast display library

A random peptide library displayed by the human MHC class I HLA-B*57:03 subtype, was used to generate the MHC class I expressing yeast display platform tested in this study. Although the AGA1 TCR is restricted by the KF11 peptide in complex with HLA-B*57:01, this TCR also binds with high affinity (Kd ~5 µM) to the near-identical HLA-B*57:03-KF11 complex (*Stewart-Jones et al., 2012*). The peptide-β2m-MHC heavy chain single chain trimer construct design was based on previously developed MHC class I yeast display scaffolds (*Adams et al., 2011*; *Birnbaum et al., 2014a*; *Gee et al., 2018*) and incorporated a single chain fusion display where HLA-B*57:03 heavy chain domains were included downstream of the KF11 epitope sequence and the β2m gene. Flexible GSSS linkers interconnected KF11, β2m ([GSSS]$_3$) and the B*57:03 heavy chain regions ([GSSS]$_4$). To minimize linker protrusions that could affect TCR binding, mutagenesis of position 84 (Tyr > Ala) in the HLA-B*57:03 alpha 1 (α1) region was performed to generate a larger linker-accommodating cavity (*Hansen et al., 2009*). A DNA construct library encoding an 11 amino acid (11mer) peptide-MHC library was generated where all peptide residues were fully randomized except for the preferred HLA-B*57:01/57:03 anchor binding residues, which were restricted to Ser, Thr and Ala at position two and to Trp, Phe and Tyr at position 11 to preserve binding to HLA-B*57:03 (*Barber et al., 1997*). All positions except for the anchor residues were generated using an NNK codon library. Position two utilized a DCK codon and position 11 used either a TGG codon for Trp or a TWY codon for Phe and Tyr. Two separate library inserts were generated to encode for the codon library NNKDCKNNKNNKNNKNNKNNKNNKNNK(TGG/TWY) with theoretical nucleotide diversities of 2.1E14 and 8.4E14 for a total diversity of 1.06E15.

The peptide-HLA-B*57:03 construct library was used to generate a yeast display library detailed above as previously described (*Gee et al., 2018*; *Birnbaum et al., 2014b*). Briefly, freshly prepared EBY-100 electrocompetent yeast were incubated with a 5:1 ratio of pMHC insert to linearized pYAL vector by mass (~1 ug vector and ~5 ug insert per electroporation) and transformed to generate a randomized peptide library. The insert contained an equal mass mixture of the polymerase chain reaction product containing the two libraries detailed above. The final library, which was generated via 20 electroporations, was then quantified as containing ~2×10$^8$ unique transformants via limiting dilution.

## Selection and sequencing of HLA-B*57:03 peptide library

Each step described below was performed essentially as previously reported and was conducted with enough yeast to ensure >10 fold coverage of the previous step's theoretical diversity. The yeast display library was seeded at an optical density of 1 (corresponding to $1 \times 10^7$ yeast/mL) for growth in SDCAA media, pH 5.5, at 30°C and grown while shaking at 250 rpm until the culture reached saturation overnight. The library was then induced in SGCAA media, pH 5.5, and allowed to incubate while shaking at 250 rpm at 20°C for 48 hr.

After induction, $1 \times 10^6$ yeast were assayed for induction by staining for a C-terminal Myc epitope tag placed between the MHC and Aga2. Next, yeast corresponding to >10 fold diversity of the initial library or the yeast from the previous round of selection were incubated with streptavidin microbeads (Miltenyi Biotec 130-048-102), washed in PBS + 0.5% BSA, and passed through a

Miltenyi LS column (Miltenyi Biotec 130-042-401) to remove non-specific binders. Yeast were then incubated with streptavidin beads co-incubated with 400 nM biotinylated TCR, washed in PBS+BSA, and then run through an LS column. Retained yeast that were then eluted from the column, washed into SDCAA, and re-cultured. Selections were repeated for an additional two rounds. Finally, a more stringent final round of selection was conducted by staining yeast with 400 nM TCR streptavidin tetramers, incubating with anti-fluorophore microbeads, and sorting via magnetic selection.

## Analysis via next-generation sequencing

Peptide enrichment and overall motifs were analyzed via next-generation sequencing as previously described. Briefly, plasmids from $5 \times 10^7$ yeast per round of selection were recovered from selected yeast (Zymoprep II kit, Zymo Research). Peptide sequences were then amplified via PCR. During the course of PCR, individual barcode sequences (to deconvolve rounds of selection) and random 8mer sequences (to increase sequence diversity) were added to the amplicons. The product of these PCRs were used in a second PCR to affix Illumina sequencing adapters. Final PCR products were gel purified, pooled, quantitated, and sequenced on an Illumina MiSeq using a v2 2 × 150 nt kit with PhiX DNA spiked in to ensure sufficient sequence diversity for high-quality reads. As detailed previously data processing, analysis, and predictions were as previously described (*Birnbaum et al., 2014b*). The forward and reverse sequencing data were used to create a single sequence of the paired-end reads using PandaSeq (*Masella et al., 2012*), deconvoluted into each round of selection by barcode, and trimmed for analysis using Geneious version 6 (Biomatters Inc). In order to correct for potential PCR or sequencing errors, sequences differing by one nucleotide from the most abundant were coalesced. Amino acid frequencies for each round were determined after translation and removal of sequences containing frame shifts or stop codons. Predictions from the profile-based search of the NCBI non-redundant protein data database (downloaded June 21, 2013) was preformed using the statistics from the third round using an amino acid frequency cut-off of 0.01. The custom scripts for data processing and analysis are available on GitHub (https://github.com/jlmendozabio/NGSpeptideprepandpred; copy archived at https://github.com/elifesciences-publications/NGSpeptideprepandpred; *Mendoza, 2020*).

## Protein production and purification

Soluble AGA1 TCR was expressed and refolded as described previously (*Boulter et al., 2003*). To facilitate stable chain pairing, an extracellular C region disulfide bridge was introduced into both the alpha and beta TCR chains by site-directed mutagenesis. The modified TCR alpha chain was cloned upstream of a hexahistidine tag and a BirA biotinylation substrate motif was inserted downstream of the modified TCR beta chain. Both chains were cloned into the PET22b+ vector (Novagen), and were expressed in the *E. coli* strain Rosetta-2 (Novagen), isolated as inclusion bodies, purified, re-solubilized, and refolded as described (*Stewart-Jones et al., 2012*; *Boulter et al., 2003*). Refolded TCR complexes were purified by anion exchange, $Ni^{+2}$ and size exclusion chromatography. The functionality of the AGA1 TCR was confirmed by cellular and SPR-based studies prior to inclusion in HLA-B*57:03 yeast display screening runs.

## Peptide generation

Conventional peptide epitopes that included a panel of library-derived peptides (Lib 1–6) and their closest non-redundant (nr) database-mined microbial sequence hits (Mic 1–6) were synthesized commercially using Fmoc chemistry (Genscript, USA). A photo-labile version of the ISPRTLNAW epitope (9MT4), where a UV sensitive 3-amino-3-(2-nitrophenyl)-propionic acid residue replaced the C-terminal penultimate Ala residue, and validated previously (*Brackenridge et al., 2011*), was sourced from LUMC, Leiden, The Netherlands.

## UV-mediated peptide exchange and sandwich ELISA assay

Refolding of HLA-B*57:01, β2m and 9MT4, and subsequent UV-photocleavage/peptide exchange were performed as outlined previously (*Brackenridge et al., 2011*; *Toebes et al., 2009*). 0.5 μM (~25 μg/mL) of UV-sensitive HLA-B*57:01-β2m-9MT4 refolded monomers were incubated with 50 μM of the various test peptides in polypropylene 96 (V-shaped) well plates (Greiner Bio-one), and the final volume was adjusted to 125 μL by adding UV exchange buffer (20mMTris pH7, 150 mM

NaCl). Samples were incubated on ice in a CAMAG UV cabinet and photo-illumination at 366 nm was performed for 60 min. Following centrifugation at 4000 g for 10 min to remove aggregated material, 90% of the sample volumes were transferred to a fresh plate, of which 3 μL was subsequently diluted 1:50 in 2% bovine serum albumin (BSA)-PBS. 50 μL of this mix was transferred (in duplicate) into an ELISA plate pre-coated overnight at 4°C with 10 μg/mL of the anti-human MHC monoclonal antibody W6/32 (Biolegend) and blocked in 2% BSA-PBS prior to sample addition. Following immobilization for 2 hr at 4°C, and extensive washing in 0.01%Tween-PBS (referred to herein as ELISA wash buffer), the plates were incubated with 50 μL of biotinylated anti-human β2-microglobulin (2M2) antibody (BioLegend, RRID:AB_493689) diluted 1:100 in 2% BSA-PBS for 1.5 hr at 4°C. Plates were washed for a further six times in ELISA wash buffer, prior to the addition of Extravidin peroxidase (Sigma) diluted 1:1000 in 2%-BSA PBS. Following 30 min at RT and a subsequent round of plate washings, 100 μL of tetramethyl benzidinesubstrate (TMB) (BioSource) was added, and plates were developed for 10 min. 100 μL of $H_2SO_4$ STOP solution (BioLegend) was added to terminate the reactions and the plates were read immediately at 450 nm absorbance using a FLUOstar OMEGA plate reader.

## Growth of microbial strains and preparation of lysates

The following microbes were grown aerobically in the specified culture conditions: Adherent-Invasive *Escherichia coli* LF82 in Luria Broth at 37°C; *Candida orthopsilosis* DSM 24508 in potato dextrose medium at 30°C; *Sporosarcina newyorkensis* DSM-23544 in Caso medium at 37°C. Strict anaerobes were cultured in a Whitley DG250 workstation at 37°C with 10% $H_2$, 10% $CO_2$, and 80% $N_2$: *Olsenella uli* DSM 7084 in DSMZ Medium 104 and *Ruminococcus gnavus* CC55_001C in filtered Reinforced Clostridial Medium (RCM). Filtered RCM was generated by centrifugation of the media at 10,000 x g for 10 min to remove agar components, followed by filtration through 0.2 μm filters. Optical density (OD600) to colony-forming unit (CFU) conversions were calculated from overnight cultures. Individual microbes were cultured for 24–48 hr in liquid media. Bacterial cells were harvested by centrifugation at 10,000 x g for 10 min, followed by three washes in sterile PBS. Cells were re-suspended in sterile PBS and heat-inactivated at 70°C for 1 hr, followed by two freeze-thaw cycles. Protein concentration was determined by BCA assay (Pierce BCA Protein Assay Kit, Thermo Fisher Scientific).

## IFNγ ELISpot assay

The standard ELISpot assay was used to detect IFNγ release by AGA1 CD8+ T cells clones, but with modifications (*Stewart-Jones et al., 2012*). 5000 peptide-pulsed HLA-B*57:01 homozygous immortalized (mycoplasma negative) B cell lines (BCLs) were incubated with 500 CD8+ T cell from two clones expressing the AGA1 TCR in the presence of 10-fold serial dilutions of KF11 and test peptides ranging from $10^{-5}$ to $10^{-9}$M final concentration. Assays were harvested following 15 hr incubation at 37°C. Results are reported as spot forming cells (SFCs) per 500 T cell input, and SFCs double that observed with media alone were considered positive.

## CD107a expression

CD107a expression following T cell stimulation was evaluated as previously described, but with slight protocol alterations (*Stewart-Jones et al., 2012*). 40,000 T cell clones were incubated with allogeneic HLA-B*57:01+ BCLs (mycoplasma negative) at a ratio of 4:1, pulsed with KF11 and the 'nr'-derived Mic peptides at final concentrations of 1 and 10 μM. Cells incubated with media alone, or in the presence of 10 ng/mL PMA were included as negative and positive controls, respectively. 5 μL of CD107a PE labelled monoclonal antibody (BD Biosciences, clone H4A3, RRID:AB_396135) was added to each well and the samples were incubated at 37°C. Golgi-Stop (BD Biosciences) was added to individual samples 1 hr post incubation at a concentration recommended by the manufacturer. Following a total of 5 hr, cells were stained with anti-human CD8 APC (BD Biosciences, clone RPA-T8, RRID:AB_398595) and fixed in BD Cell Fix prior to flow cytometry analysis using a CyAn. Data analysis was performed using FlowJo software.

## T cell functional assays using bacterial cell lysates

$2 \times 10^6$ HL60 cells (ATCC CLL-240/RRID:CVCL_0002, STR profiled and mycoplasma negative) were treated overnight with calcium ionophore and IFNγ at 37°C to induce DC/myeloid-like differentiation and to enhance MHC class I expression (*Wu et al., 2004a*; *Wu et al., 2004b*). The cells were subsequently divided into 48 well NUNC plates and 20 μg/mL of the test bacterial cell lysates were added to individual wells. Following a further 7 hr of incubation, the cells were washed extensively and their ability to stimulate AGA1-expressing T cell clones was assessed by ELISpot assay. 15,000 lysate-pulsed HL60 cells were incubated with 500 CD8+ T cell from two clones expressing the AGA1 TCR for 15 hr prior to development. Positive control included non-lysate-pulsed HL60 and T cells incubated with 10ng/mL PMA. Results are reported as spot forming cells (SFCs) per 500 T cell input, and SFCs double that observed with media alone were considered positive.

## Acknowledgements

The authors thank the support staff at both Stanford and Oxford University for their technical assistance. We also thank Dr. Dris Elatmioui at LUMC, The Netherlands who synthesized the UV-sensitive peptide. This work was supported by the following grants: MRC_MR/M019837/1(GMG), NIH 5R01AI103867 (KCG), U19AI057229 (KCG) and HHMI (KCG).

## Additional information

### Competing interests

Marvin H Gee: MHG is a co-founder of 3T Biosciences. K Christopher Garcia: KCG is a co-founder of 3T Biosciences. The other authors declare that no competing interests exist.

### Funding

| Funder | Grant reference number | Author |
| --- | --- | --- |
| Medical Research Council | MR/M019837/1 | Geraldine Martina Gillespie |
| National Institutes of Health | 5R01AI103867 | K Christopher Garcia |
| National Institutes of Health | U19AI057229 | K Christopher Garcia |
| Howard Hughes Medical Institute | | K Christopher Garcia |

The funders had no role in study design, data collection and interpretation, or the decision to submit the work for publication.

### Author contributions

Juan L Mendoza, Marvin H Gee, Data curation, Software, Formal analysis, Validation, Investigation, Methodology, Writing - original draft, Writing - review and editing; Suzanne Fischer, Data curation, Software, Formal analysis, Validation, Investigation, Methodology; Lilian H Lam, Methodology, Writing - review and editing; Simon Brackenridge, Fiona M Powrie, Methodology; Michael Birnbaum, Conceptualization, Resources, Software, Formal analysis, Validation, Investigation, Methodology, Writing - review and editing; Andrew J McMichael, Conceptualization, Investigation, Writing - original draft, Writing - review and editing; K Christopher Garcia, Funding acquisition, Conceptualization, Data curation, Investigation, Methodology, Writing - review and editing; Geraldine M Gillespie, Conceptualization, Data curation, Investigation, Methodology, Writing - original draft, Writing - review and editing

### Author ORCIDs

Lilian H Lam http://orcid.org/0000-0003-3582-9209
Simon Brackenridge https://orcid.org/0000-0002-0587-7560
K Christopher Garcia https://orcid.org/0000-0001-9273-0278
Geraldine M Gillespie https://orcid.org/0000-0002-1075-870X

Decision letter and Author response
Decision letter https://doi.org/10.7554/eLife.58128.sa1
Author response https://doi.org/10.7554/eLife.58128.sa2

## Additional files

### Supplementary files

• Supplementary file 1. Non-redundant ('nr') database recovered KF11-related peptides, TCR sequence similarity of T cell clones and HLA-B*57:01 HdH peptide binding hierarchies. Table 1: Round 3 of the AGA-1 TCR yeast display selection results were used to identify sequence related peptides from the non-redundant ('nr') database. Listed are the prediction results for KF11-related peptides derived from the gag HIV protein. GI identification and blast scores are included. Table 2: AGA-1 TCR yeast display selection results from Round 3 screens were used to identify sequence related peptides from the non-redundant ('nr') database. Listed are the prediction results of sequence-related, non-KF11 peptides. GI identification, blast scores, % identity to the KF11 peptide and to 'nr' database mined peptide hits are included. Table 3: TCR alpha chain amino acid sequence identity of clone 1.2 and the AGA1 (clone 1.1) TCR. Clones 1.1 and 1.2 both utilize the V alpha 5 (AV5) chain segments, encode identical CDR3 motifs and carry one amino acid sequences difference that maps to the J alpha (AJ) region (underlined). This residue is outside the TCR:peptide-MHC binding interface described previously for the AGA1 TCR-B*57:03-KF11 co-complex (red) (*Stewart-Jones et al., 2012*). AGA1 TCR clone 1.1 and the related clone 1.2 are 100% sequence identical across the CDR3 and J regions of their V beta19 (BV19) TCR chain sequences (CASTGSY-GYTFGSGTRLTVT) (*Stewart-Jones et al., 2012*). The TCR V region, CDR3 and J region boundaries as defined by ImMunoGeneTics (IMGT), http://www.imgt.org/IMGTrepertoire/. Table 4: Peptide binding strength hierarchy ranked on the basis of UV exchange HLA-B*57:01 peptide binding ELISA data. The index KF11 epitope and the HdH peptides are ranked in descending order according to their peptide binding strength as determined using the UV exchange peptide binding assay. High, medium and low binders are color-coded from dark to light shades of grey, respectively. Amino acids differences specific to low/medium binders outside the p4-p6 TCR recognition interface are highlighted (bold squared). p1 = peptide position 1, etc.

• Source data 1. Library peptide-based returned 'nr' hits_KF11 variant peptides.

• Source data 2. Library peptide-based returned 'nr' hits_non-KF11 related peptides.

• Source data 3. Correlation between ELISA data and NetMHC Pan4.1 binding predictions for HdH peptides.

• Transparent reporting form

### Data availability

Next-generation sequencing data has been deposited in the Sequence Read Archive under accession numbers SAMN14376837 and SAMN14376838. Accession Sample Name SPUID Organism Tax ID Strain SAMN14376837 AGA1_255 AGA1_255 synthetic construct 32630 EBY100 SAMN14376838 AGA1_177 AGA1_177 synthetic construct 32630 EBY100 https://www.ncbi.nlm.nih.gov/biosample/14376837 https://www.ncbi.nlm.nih.gov/biosample/14376838.

The following datasets were generated:

| Author(s) | Year | Dataset title | Dataset URL | Database and Identifier |
|---|---|---|---|---|
| Gee M | 2020 | BioSample: SAMN14376837; Sample name: AGA1_255; SRA: SRS6329182 | https://www.ncbi.nlm.nih.gov/biosample/14376837 | NCBI BioSample, SAMN14376837 |
| Gee M | 2020 | BioSample: SAMN14376838; Sample name: AGA1_177; SRA: SRS6329183 | https://www.ncbi.nlm.nih.gov/biosample/14376838 | NCBI BioSample, SAMN14376838 |

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
