## [Decision Letter]

**Acceptance summary:**

The reviewers are impressed with this manuscript, which is well-written and includes rigorous experiments. The central issue addressed by the study holds interest for the broader community. We believe that TCR cross-reactivity is informative for understanding the broader implications of adaptive immunity. It is especially interesting that the presented peptide, KAFSPEVIPMF, is not in the set of top peptides enriched after sorting, suggesting that pMHC/TCR affinity is not the whole story of T cell activation.

**Decision letter after peer review:**

Thank you for submitting your article "MHC yeast display-interrogation of a conserved, HIV specific TCR reveals cross-reactivity to bacteria-diverse peptides" for consideration by *eLife*. Your article has been reviewed by three peer reviewers, and the evaluation has been overseen by a Reviewing Editor and Satyajit Rath as the Senior Editor. The following individuals involved in review of your submission have agreed to reveal their identity: Brian Baker (Reviewer #2); David Kranz (Reviewer #3).

The reviewers have discussed the reviews with one another and the Reviewing Editor has drafted this decision to help you prepare a revised submission.

Summary:

Fischer et al. reports fascinating results from a technical tour-de-force studying TCR cross-reactivity. The paper as written is concise, clearly presented, and, while dense, is approachable. Materials and methods are detailed appropriately, experiments are properly executed, and results are correctly interpreted. Detailing TCR cross-reactivity in this way is informative and the concluding speculations are intriguing. An interesting observation is that the presented peptide, KAFSPEVIPMF, is not in the set of top peptides enriched after sorting, confirming the nuance that pMHC/TCR affinity is not the whole story of T cell activation.

Overall, the consensus of the reviewers was that the manuscript is well-written and experiments are rigorously performed, demonstrating considerable technical finesse. The central issue addressed by the study holds interest for the broader community. While generating support for publication in *eLife*, the reviewers thought that revising the manuscript would provide needed additional detail and dial back some more speculative conclusions and discussion. Additional experiments are not deemed necessary. Important points meriting additional discussion and consideration include the limitations of a yeast-display library that covers only a tiny percentage of the possible sequence space and the representation of the eliciting KF11 peptide in the library and its apparent failure to enrich. Also, while quite intriguing, the synthesis of the presented results do not fully support the broad conclusion stated in the title: "MHC yeast display-interrogation of a conserved, HIV specific TCR reveals cross-reactivity to bacteria-diverse peptides". Confirming the microbiota connection would require additional experimental scope. Therefore, the reviewers feel that a more conservative tone in the Discussion should be embraced in revision.

1) On the note above, do p-MHC binding affinity *predictions* (e.g., NetMHC or similar) track with the data? Given the results in Figure 3B, can the authors comment on how incorporating MHC binding affinity predictions into the database scanning would add to the process?

2) Subsection “Cross-reactive potential of AGA1+ T cell clones to library- and database-derived peptide hits.” – clone 1.2. How homologous is this TCR? This should be somewhere, to include loop sequences. Is there anything to be learned regarding identity/homology in the sequences and responses to the peptides?

3) The first involves the statistical probability of properly identifying cross-reactive peptides from a library where the sequence-space that is screened is only one in over a million (i.e. library of 10e8, where theoretical diversity is 10e15). Thus, most cross-reacting peptides are likely missed. In fact, one presumes that they did not even isolate the wild type FK11 or its variants and there needs to be a discussion of this.

4) In a similar way, the authors need to discuss whether they used the FK11 sequence to search the microbial database and what they identified in this search.

5) A significant concern is whether identifying the Mic 1/HdH sequences as cross-reacting ligands for Aga1 TCR from their search has any physiological significance. Or, is it just the likelihood of finding peptides that cross-react due to the now well understood principles that a single TCR cross-reacts with many peptide ligands. *S. newyorkensis* is not an intracellular human pathogen, nor a known commensal organism. To have functional significance one would imagine very large burdens would need to be encountered in order to have sufficient peptide presented through APC uptake, processing, and presentation by HLA-B57. The experiment with lysates from *S. newyorkensis* tries to address this, but it is not even clear that the HdH peptide is the actual ligand in that experiment. While it will be experimentally very difficult to prove that *S. newyorkensis* has an influence in vivo on T cells that bear Aga1-related TCRs, it would at least be useful to know that the stimulating ligand in the lysate was indeed from HdH. If HdH cannot be deleted, perhaps it could be introduced into another “non-inducing” bacterial host to show lysates of that stimulate Aga1 T cells.

---

## [Author Response]

Overall, the consensus of the reviewers was that the manuscript is well-written and experiments are rigorously performed, demonstrating considerable technical finesse. The central issue addressed by the study holds interest for the broader community. While generating support for publication in eLife, the reviewers thought that revising the manuscript would provide needed additional detail and dial back some more speculative conclusions and discussion. Additional experiments are not deemed necessary. Important points meriting additional discussion and consideration include the limitations of a yeast-display library that covers only a tiny percentage of the possible sequence space and the representation of the eliciting KF11 peptide in the library and its apparent failure to enrich. Also, while quite intriguing, the synthesis of the presented results do not fully support the broad conclusion stated in the title: "MHC yeast display-interrogation of a conserved, HIV specific TCR reveals cross-reactivity to bacteria-diverse peptides". Confirming the microbiota connection would require additional experimental scope. Therefore, the reviewers feel that a more conservative tone in the Discussion should be embraced in revision.

In light of the these comments, we have moderated the Discussion as requested. Although we have left some of the speculative discussion in place, we do specifically state, on two occasions in this amended draft, that we have not shown in vivo evidence that the AGA1 TCR is stimulated by the HLA-B*57:01 restricted bacterial-derived HdH peptides (Discussion: paragraph five and six). We have also adjusted the title to reflect that peptides which are sequence identical to microbial peptides are recognised by the TCR to help state more accurate claims.

Revisions for this paper:1) On the note above, do p-MHC binding affinity predictions (e.g., NetMHC or similar) track with the data? Given the results in Figure 3B, can the authors comment on how incorporating MHC binding affinity predictions into the database scanning would add to the process?

The reviewer raises an interesting point that we have now addresses in the revised manuscript. We have added the specific section to Results, subsection “Recognition HdH peptides by AGA1-expressing T cell clones” paragraph one. There is a negative, albeit very weak (R2= -0.18), correlation between NetMHCPan4.1 (http://www.cbs.dtu.dk/services/NetMHCpan/) predicted binding affinities for KF11 and bacterial HdH peptides and the ELISA-generated binding data (Figure 4B, Figure 4—figure supplement 1). One factor that is likely to confound the strength of this correlation is a lack of data generated using either in vitro binding assays for HLA-B*57 and/or from mass spectrometry-based peptide elution studies for binding of longer peptides to MHC, and this would potentially limit the predictive power of NetMHC when such peptides are considered. As an aside to this, we have also ranked the HdH and KF11 peptides according to the strength of peptide binding ELISA data and patterns are evident. The strongest HdH binders carry in their peptide sequences: p2 Thr/Ser (preferred anchors), p3Iso, p7Iso and a Pro as p8 or 9 (but not both) and p10Try. Peptides in the medium and low binding categories although containing some combinations of these residues have noted single amino acid changes and/or combinations of changes in these positions that could modulate binding strength. Supplementary file 1–table 4 has been added to summarize this.

2) Subsection “Cross-reactive potential of AGA1+ T cell clones to library- and database-derived peptide hits.” – clone 1.2. How homologous is this TCR? This should be somewhere, to include loop sequences. Is there anything to be learned regarding identity/homology in the sequences and responses to the peptides?

We apologise for the omission of this information. The relevant sequences have now been added to the manuscript (Supplementary file 1–Table 3) and is now referenced in the Results section (subsection “Cross-reactive potential of AGA1+ T cell clones to library- and database-derived peptide hits.”, last paragraph). The Vbeta19 chains of the AGA1 TCR, which corresponds to clone 1.1, is 100% sequence identical across the CDR3 and J region of the V beta 19 chain expressed by clone 1.2. One amino acid difference distinguishes the TCR alpha chains of AGA1 (clone 1.1) and clone 1.2 – this maps to within the J region (underlined).

This residue is outside the peptide-MHC binding interface described previously for the AGA1-B*57:03-KF11 co-complex (noted here in italics). Although we cannot exclude the possibility that this change causes subtle differences to the recognition of these peptides, the functional data we generated in vitro using clones 1.1 and 1.2 did not suggest any discernible differences in patterns of HdH peptide recognition.

3) The first involves the statistical probability of properly identifying cross-reactive peptides from a library where the sequence-space that is screened is only one in over a million (i.e. library of 10e8, where theoretical diversity is 10e15). Thus, most cross-reacting peptides are likely missed. In fact, one presumes that they did not even isolate the wild type FK11 or its variants and there needs to be a discussion of this.

It is true that the functional library cannot fully cover the theoretical diversity of cross-reactive peptides. Although the library design generated here allowed for KF11 to be present, it was not found in either the naive library nor in subsequent rounds of AGA1 TCR-enriched libraries. Hence, the yeast clone was probably not present as a transformant in this study. However, the strength of this method is that the restricting peptide ligand doesn’t have to be in the library – using algorithms and statistics, it is possible to pull out at the restricting peptides, in addition to related peptides, from available databases. This is exactly what we observed for the KF11 peptide which emerged as the top database-mined peptide hit using library peptide-led database searches. We have added a discussion to this effect (Discussion paragraph three).

4) In a similar way, the authors need to discuss whether they used the FK11 sequence to search the microbial database and what they identified in this search.

It is true that simple algorithms based on KF11 peptide could have informed hunts for additional peptide ligands. However, as KF11 and the HdH peptides share only 50% sequence similarity, it’s unlikely that they would have been mined using more simplistic searches based on the sequence of KF11 alone. The MHC-yeast display/TCR-informed database searches enables more focused and definitive screens given that algorithms generated are data-led and take into account the broad range of peptides that can physically bind the MHC and that are also physically recognised by the TCR.

5) A significant concern is whether identifying the Mic 1/HdH sequences as cross-reacting ligands for Aga1 TCR from their search has any physiological significance. Or, is it just the likelihood of finding peptides that cross-react due to the now well understood principles that a single TCR cross-reacts with many peptide ligands. S. newyorkensis is not an intracellular human pathogen, nor a known commensal organism. To have functional significance one would imagine very large burdens would need to be encountered in order to have sufficient peptide presented through APC uptake, processing, and presentation by HLA-B57.

We accept the reviewers comments and can only speculate as to whether these bacterial antigens can elicit MHC restricted T cell cross-reactivity in vivo. Although *S. newyorkensis* is not a commensal, nor is it an intracellular pathogen, this microbe has been reported as a pathogen that can cause infection in humans, and there is also potential for exposure via environmental contamination, especially from outdoor terrains, most notably soil. Also, as pointed out in the manuscript, we identified and tested a number of additional HdH peptides from other Bacilli, and a variety of peptides from genera/species other than *S. newyorkensis* were recognised by the AGA1 TCR. Most interestingly, and included in this group, was a HdH peptide derived from a commensal known as *Bacillus massiliosenegalensis*, raising the likelihood that at least one population of microbiome commensals carry a HdH peptide in vivo.

The experiment with lysates from S. newyorkensis tries to address this, but it is not even clear that the HdH peptide is the actual ligand in that experiment. While it will be experimentally very difficult to prove that S. newyorkensis has an influence in vivo on T cells that bear Aga1-related TCRs, it would at least be useful to know that the stimulating ligand in the lysate was indeed from HdH. If HdH cannot be deleted, perhaps it could be introduced into another “non-inducing” bacterial host to show lysates of that stimulate Aga1 T cells.

We want to assure the reviewer that the lysates from *S. newyorkensis* comprised the only bacterial lysates that was well recognised by the AGA1 TCR – this observation was reproducible during different rounds of biological repeats and for clones 1.1 and 1.2. However, we do accept that recognition of the lysate requires more detailed validation. We will endeavour to address the nature of the ligand once the post-COVID restrictions allow us to pursue this work.